# Minimally Invasive Approach to Gastric GISTs: Analysis of a Multicenter Robotic and Laparoscopic Experience with Literature Review

**DOI:** 10.3390/cancers13174351

**Published:** 2021-08-27

**Authors:** Graziano Ceccarelli, Gianluca Costa, Michele De Rosa, Massimo Codacci Pisanelli, Barbara Frezza, Marco De Prizio, Ilaria Bravi, Andrea Scacchi, Gaetano Gallo, Bruno Amato, Walter Bugiantella, Piergiorgio Tacchi, Alberto Bartoli, Alberto Patriti, Micaela Cappuccio, Klara Komici, Lorenzo Mariani, Pasquale Avella, Aldo Rocca

**Affiliations:** 1General Surgery Department, ASL 2 Umbria, San Giovanni Battista Hospital, 06034 Foligno, Italy; g.cecca2003@libero.it (G.C.); gianlucacostaphd@gmail.com (G.C.); walterbugiantella@alice.it (W.B.); piergiorgio.tacchi@uslumbria2.it (P.T.); alberto.bartoli@uslumbria.it (A.B.); albertopatriti@gmail.com (A.P.); lorenzo.mariani@uslumbria.it (L.M.); 2General, Minimally Invasive and Robotic Surgery, Department of Surgery, ASL 2 Umbria, San Matteo Hospital, 06049 Spoleto, Italy; 3General Surgery Unit, San Donato Hospital, 52100 Arezzo, Italy; frezzabarbara@libero.it (B.F.); deprizio.marco@gmail.com (M.D.P.); 4Surgery Center, University Campus Bio-Medico of Rome, 00128 Rome, Italy; 5Department of General Surgery, San Giovanni Battista Hospital, 06034 Perugia, Italy; morrisey1759@hotmail.it; 6UOC General Surgery and Laparoscopic Surgery, Department of Surgery P. Valdoni, Policlinic Umberto I, Sapienza University of Study of Rome, 00161 Rome, Italy; massimo.codacci@gmail.com; 7Histopathology Department, Usl Umbria 2, San Giovanni Battista Hospital, 06034 Foligno, Italy; ilaria.bravi@uslumbria2.it; 8Department of Medicine and Health Sciences V. Tiberio, University of Molise, 86100 Campobasso, Italy; scacchiandrea@me.com (A.S.); micaelacappuccio24@gmail.com (M.C.); klara.komici@unimol.it (K.K.); avella.p@libero.it (P.A.); 9Department of Medical and Surgical Sciences, University of Catanzaro, 88100 Catanzaro, Italy; gaetanogallo1988@gmail.com; 10Department of Colorectal Surgery, S. Rita Clinic, 13100 Vercelli, Italy; 11Department of Clinical Medicine and Surgery, University of Naples Federico II, 80126 Naples, Italy; bramato@unina.it; 12Division of General Surgery, Ospedali Riuniti Marche Nord, 61121 Pesaro, Italy

**Keywords:** gastrointestinal stromal tumor, GISTs, robotic surgery, laparoscopic surgery, gastric resection, minimally invasive surgery, intracorporeal suture

## Abstract

**Simple Summary:**

Gastrointestinal stromal tumors (GISTs) represent about 1–3% of all gastrointestinal malignancies, of which 50–60% are gastric GISTs (GGs). To the date, surgery represents the best therapeutic option, and the robotic gastric surgery could gain an important role, overcoming many laparoscopic drawbacks. The aim of this study is to evaluate safety and effectiveness of minimally invasive surgery (MIS) for GGs, reporting 10-year experience of three different centers. We included a population of 81 patients who underwent MIS approaches (36 laparoscopy vs. 45 robotic surgery). Seventy-two (72) patients were enrolled in a follow-up program to evaluate the long-term oncological outcomes. Furthermore, we discussed some technical notes and also we analyzed the operative and peri-operative outcomes. In conclusion, our results suggest that the robotic approach might be a suitable treatment, especially for GISTs >5 cm, even located in unfavorable places, despite longer operative time and costs than laparoscopic approach.

**Abstract:**

**Background**: Gastrointestinal stromal tumors (GISTs) are most frequently located in the stomach. In the setting of a multidisciplinary approach, surgery represents the best therapeutic option, consisting mainly in a wedge gastric resection. (1) Materials and methods: Between January 2010 to September 2020, 105 patients with a primary gastrointestinal stromal tumor (GISTs) located in the stomach, underwent surgery at three surgical units. (2) Results: A multi-institutional analysis of minimally invasive series including 81 cases (36 laparoscopic and 45 robotic) from 3 referral centers was performed. Males were 35 (43.2%), the average age was 66.64 years old. ASA score ≥3 was 6 (13.3%) in the RS and 4 (11.1%) in the LS and the average tumor size was 4.4 cm. Most of the procedures were wedge resections (*N* = 76; 93.8%) and the main operative time was 151 min in the RS and 97 min in the LS. Conversion was necessary in five cases (6.2%). (3) Conclusions: Minimal invasive approaches for gastric GISTs performed in selected patients and experienced centers are safe. A robotic approach represents a useful option, especially for GISTs that are more than 5 cm, even located in unfavorable places.

## 1. Introduction

Gastrointestinal stromal tumors (GISTs) are the most common intestinal mesenchymal tumors [1,2,3], with an estimated incidence of 1–2 cases per 100,000 (one hundred thousand) people a year. They represent about 1–3% of all gastrointestinal malignancies [4]. Fifty–sixty percent of cases are represented by gastric GISTs that are mostly located in corpus and fundus [5,6,7,8,9].

Gastric GISTs (GGs) are generally asymptomatic and are incidentally discovered during endoscopic or radiologic exams. When they are symptomatic, clinical findings usually range from abdominal pain to massive gastrointestinal bleeding due to tumor ulceration [5,10,11].

The diagnostic work up provides endoscopy with biopsy, abdominal ultrasound, and computed tomography (CT). Magnetic resonance imaging (MRI) and endoscopic ultrasound (EUS) with fine needle biopsy could be helpful in selected cases when differential diagnosis is uncertain.

GGs show two different patterns of growth in the gastric cavity: intraluminal or exophytic. The important survival prognostic factors that pathologists describe are: tumor size and location, eventual tumor rupture, mitotic rate, c-KIT expression, and its mutation [12].

A predicted risk score of recurrence after GGs resection is stated in the study written by Fletcher et al. which was essentially based on tumor size and mitotic count [13].

Surgical resection represents the treatment of choice in primary localized GGs, except for small tumors (less than 2 cm), which might be monitored or addressed by endoscopic resection [14]. The main goals of surgical resection could be achieved through R0 gastric wedge resections and to avoid an intraoperative tumor rupture, while lymphadenectomy of clinically negative nodes is not indicated [15].

Open surgery has been the standard care for a long time and it is still the best option for large resectable tumors [16].

Recently, laparoscopic resections have become more widely adopted in upper gastro-intestinal (GI) surgery, for GISTs or for other gastric tumors [17]. Concerning GISTs, the minimally invasive (MI) approach might be influenced by tumor dimension and location surrounding organs infiltration and surgeon’s technical laparoscopic skills and experience. Besides these findings, the risk of tumor rupture still represents a major concern for the laparoscopic resection.

More recently, robotic gastric surgery has gained an important role in abdominal surgery, overcoming many laparoscopic drawbacks [18,19,20,21,22,23,24,25]. The aim of this study is to evaluate safety and effectiveness of the MI approach for GGs reporting 10-year experience of three different centers, which are trained in robotic and laparoscopic GGs resection.

## 2. Methods

### 2.1. Study Design

This was a retrospective multicenter study developed according to the Strengthening the Reporting of Observational Studies in Epidemiology (STROBE) statement for cohort studies [26].

### 2.2. Study Population

We retrospectively reviewed the prospectively maintained database of gastric surgery of three different Italian Surgical Oncology Units (Umbria2 Local Health Service Hospitals San Giovanni Battista in Foligno along with San Matteo in Spoleto, and Toscana Sud-Est Health Service Hospital San Donato in Arezzo).

Medical charts of patients who underwent MI gastric resection for GIST from January 2010 to September 2020 were reviewed. Only procedures performed or supervised by senior staff surgeons qualified in laparoscopic and robotic upper-GI surgery were considered [12]. We further selected only GISTs confirmed by pathological examination. Exclusion criteria included open approach, duodenal GIST diagnosis, and resections performed in emergency settings.

We excluded the procedures that were not performed or tutored by experienced surgeons (defined as surgeons who have performed at least one-hundred elective or emergency MI Upper GI procedures) due to potential bias.

We also excluded patients affected by pre-operative diagnosis of Stage IV disease and any other finding of adjacent organs involvement requiring associated multi-organ resections or neo-adjuvant chemotherapy (Imatinib). A multidisciplinary team (gastro-intestinal tumor board), composed of oncologists, radiologists, gastroenterologists, and an upper-GI oncologist surgeon, evaluated all cases and decided case by case the best treatment options.

All patients classified as high risk and selected cases in medium risk class received adjuvant treatment after oncological evaluation.

All patients signed a proper informed consent for the scientific anonymous use of clinical data. The study was conducted according to the guidelines of the Declaration of Helsinki and approved by the Institutional Review Board of University of Molise (protocol number 10/21, approved date: 12 May 2021).

### 2.3. Variables and Definitions

The collected data included: demographic characteristics such as age, sex, and body mass index (BMI), (BMI 30 corresponds to Class I obesity); the American Society of Anaesthesiologists (ASA) score; preoperative comorbidities classified according to the Charlson comorbidity index (CCI) [27]; clinical presentations (bleeding, anemia, etc.) or eventual incidental diagnosis; preoperative investigations (endoscopy with biopsy, ultrasonography, CT scan, MRI scan, endoscopic ultrasound-guided fine-needle aspiration).

Tumor characteristics were registered and analyzed considering size, gastric location, and type of growth (endophitic/exophitic). The tumors’ size was defined as the lesion maximum diameter at pathological examination. As regards to size, the MI approach was adopted following the guidelines available at the time of surgery. It is important to note that indications changed during the 10-year period of the study.

We reviewed the type of surgical procedure (wedge or formal gastric resection), operative time, intraoperative mean blood loss and peri-operative blood transfusions, associated abdominal surgery, conversion to open surgery, time to oral intake, post-operative length of hospital stay, intra-operative and post-operative complications (according to Clavien-Dindo score) [28], prognostic information including R0 or R1 margins, and intraoperative tumor rupture.

Complications according to Clavien-Dindo classification [28], readmissions, and mortality were collected up to 90 days after surgery.

Tumor risk recurrence was calculated according to Fletcher score [4,29] and the data are listed in Table 1.

Histopathological data included immunohistochemical analysis performed using markers such as CD117, CD34, (only positive were considered), SMA, and S-100 protein. The mitotic index was measured through the HPF.

In our database we registered patients dividing high and low mitotic index. The cutoff point was defined as five of more mitoses registered at 50 HPF (Table 1).

### 2.4. The Tumor Location

According to Arseneaux et al., the tumor localizations include the following: anterior gastric wall and greater curvature, posterior wall, lesser curvature, esophageal junction, and antro-pyloric region [30].

We divided the tumor localizations into favorable and less favorable resections for an MI approach [30] (Figure 1 and Table 2).

We also registered the three different patterns of development and growth of GGs: mainly endoluminal, exophytic, or transmural (Figure 2).

### 2.5. Follow up Program

Follow-up data collection was performed through telephone interview and/or regular outpatient visits, with CT scan and gastroscopy 6 months after surgery [31].

Following the Italian Oncological Guidelines on GIST [31], the date of the follow-up management after GGs resection is not clearly defined in the guidelines [31]. The standard follow-up provides abdominal CT-scan each 6 months during adjuvant chemotherapy (Imatinib) then each 3–4 months for 2 years and then 1–2 times/year for 10 years. For low grade GIST, in absence of clear indications, we performed CT-scan 1 time/year for 5 years.

PET-CT was scheduled after eventual Tyrosine kinase inhibitor (or similar) therapy or in case of recurrence. Because no death was related to GIST, the cause-specific survival (CSS) was not evaluated. There were censored data regarding patients who died from other causes, who were alive on date of their last follow-up, or were lost. Only disease-free survival (DFS) was calculated. DFS was defined as the period from surgery to recurrence.

### 2.6. Technical Notes

#### 2.6.1. Laparoscopic Approach

Under general anesthesia, the patient is placed in supine reverse-Trendelenburg position (approximately 20°).

Surgeon stood between the patient’s legs. Interventions are performed using four/five trocars, as depicted in the Figure 3. Pneumoperitoneum is induced using Veress needle in the left upper quadrant (Palmer point), and maintained at 12 mm Hg abdominal pressure. The abdominal cavity is first inspected to assess the operability.

Wedge resections are routinely performed using a laparoscopic linear stapler, especially for tumors located in favorable sites (anterior, posterior wall, and greater curvature), with or without a reinforcing running suture on the resection line. R0 margin-free resection and the risk of tumor rupture are the main pitfalls to pay attention to during laparoscopic surgery. In all cases the tumor specimen extraction should be performed using an endoscopic bag, in order to avoid spillage and abdominal wall contamination. We extract the specimen using a trocar site enlargement or Pfannenstiel incision for large tumors. The nasogastric tube placed during the operation was generally removed the day after surgery.

#### 2.6.2. Robotic-Assisted Surgery

We used daVinci Robot System Si (Intuitive Surgical Inc., Sunnyvale, CA) from 2010–2017, then the new Da Vinci Xi platform became available. Only two of the three centers enrolled in the study performed robotic resections. The general rules adopted in laparoscopy are also observed with the robotic approach, including patient positioning. The main differences involve the device docking, being the last da Vinci type (Xi) more versatile and allowing a better ergonomics, with a consequent easier and faster docking. The robotic arms come from the patient’s head. We use four robotic ports, one placed just above the umbilicus for the 30° camera, and the others positioned as depicted in Figure 3.

A 5th accessory trocar for the assistant (slightly below the port-line) is placed in the left half of the abdomen. We commonly use a monopolar curved scissors and fenestrated bipolar and prograsp forceps for retraction; sutures are performed using a robotic articulated needle-driver. The intracorporeal anastomosis consists of a manual two layers running suture to close the gastric wall defect. In more detail, we performed a long-term absorbable 2–0 suture or a single barbed suture with a back-and-forth technique (Figure 4, Figure 5, Figure 6 and Figure 7). During robotic operations we do not use energy devices for dissection nor an endoscopic stapler for wedge gastric resections. These devices are reserved for standard gastrectomies. The use of Indocyanine green (ICG) technique during gastric resection to better identify the tumor was performed in 12 cases over 47 (Figure 4 and Figure 6). Post-operative work-up is the same for both techniques.

An intraoperative upper endoscopy was performed in 31 cases (38.3%) either to define the exact tumor location in completely endophytic GISTs or to check sutures after gastric wall reconstructions. In five (6.2%) cases an endoscopic intraoperative ultrasound was performed for endophytic lesion identification.

### 2.7. Statistical Analysis

Dichotomous data and counts were presented in frequencies, whereas continuous data were presented as mean values ± standard deviations (SD) and/or median with 25–75 interquartile range (IQR) and minimum-maximum range. To compare differences in frequencies, Fisher’s exact test or χ^2^ test with or without Yates’ correction were performed.

Differences between means were compared using the Mann–Whitney U test. Receiver operating characteristic (ROC) curve analysis was performed to calculate the cut-off value corresponding to the maximum Youden’s index of any relevant continuous variable in order to transform it in a dichotomous one. Survival data were analyzed using Kaplan-Meier method, and log-rank test was used to compare curves. Univariate and multivariate Cox regression analysis was carried out to identify hazard ratio (HR) of factors associated with time to recurrence. All variables with *p* value < 0.10 at univariate analysis were entered into a multivariate model. A *p*-value ≤ 0.05 was considered statistically significant. Statistical analysis was carried out using StataCorp2019 STATA Statistical Software: release 16 (StataCorp LLC, College Station, TX, USA).

## 3. Results

During the study period, a total of one hundred and five (105) consecutive patients underwent surgical resection for primary GGs confirmed at immunohistochemical finding, including open resections, LS and RS.

Eighty-one (81) resections fulfilled the inclusion criteria and were therefore approached with MI surgery. Nine patients were lost at FU. Therefore, seventy-two (72) patients were included in the study: thirty-six (36) patients underwent conventional laparoscopic surgery (LS) and forty-five (45) underwent robot-assisted surgery (RS).

Patients’ baseline characteristics and clinic-pathologic data of tumors are summarized in Table 2 and Figure 8.

Mean age of population was 66.64 years old (range: 35–87 yrs), 12 patients (14.8%) were very elderly (>80 yrs), 9 of them underwent RS, and 3 underwent LS.

Obese patients with a BMI >30 were 9 (11.1%), 6 in the RS (13.3%), and 3 in the LS (8.3%).

Thirty-three patients (40.75%) were affected by one or more comorbidities at the time of surgery (Table 2). No patient in the MI series underwent neoadjuvant chemotherapy (Imatinib).

Regarding clinical debut we observed abdominal pain (including simple discomfort) in 39.5% (32) of patients and gastric bleeding or anemia in 28.4% (23) (Table 2). In 53% of cases patients were asymptomatic and the diagnosis was incidental. In two obese patients undergoing a robotic gastric by-pass and a laparoscopic sleeve gastrectomy, the GGs was an intraoperative incidental finding. Those two patients showed a small GISTs with tumor diameter ≤20 mm.

Mean tumor size was 4.4 cm (1.5–12 cm), 5.1 cm in the RS, and 3.7 cm in the LS respectively (*p* = 0.0078). Tumors larger than 5 cm were 18 (22.2%), 15 in the RS series (33.3%), and 3 in the LS (8.3%) respectively (Table 2).

Concerning GGs location, “unfavorable locations” (juxtacardial, lesser curvature and antro-pyloric) represented the 35.8% of the total: 8 in the juxtacardial area, 9 in the lesser curvature, and 12 in the antro-pyloric region [30].

The majority of unfavorable resections (68.9% vs. 31.1%) were found in the RS.

In total, 35.8% of tumors had an endophytic growth, while 64.2% showed an exophytic or transmural pattern.

We performed 76 (93.8%) gastric wedge resections, and only 5 distal gastrectomies, 2 (4.5%) in RS and 3 (8.4%) in LS (*p* = 0.6511) (Table 3).

The overall conversion rate to open surgery was 6.2% (5 cases), 3 cases (8.4%) in the LS, and 2 cases (4.5%) in the RS (*p* = 0.6511).

In the RS conversion was required in one case for oncological reasons (a large 10-cm tumor infiltrating the spleen), and in a second case for a 7-cm GIST in a patient already resected for peptic ulcer (adhesions and risks of tumor rupture addressed to a safer open procedure). In the LS conversions were due to an intraoperative bleeding and to better manage two large lesions located in unfavorable sites with high risk of rupture (in one case the capsule rupture occurred). Finally, we report only one case of hand-assisted anastomosis during laparoscopic approach to gastric resection.

Associated major abdominal procedures were performed in 15 cases (18.5%), 10 (22.2%) in the RS vs. 5 (13.8%) in the LS respectively (*p* = 0.26). Associated procedures are listed in more detail in Table 3.

No statistical differences were found in the two groups concerning associated surgical procedures.

Overall mean operative time was significantly higher during RS versus LS (*p* = 0.0002).

The hospital stay did not differ between the two series (6.7 in the RS vs. 6.3 in the LS) (*p* = 0.39).

The 30-day readmission rate was null.

No significantly differences between the two series were observed about time to return to bowel function and oral intake (Table 3). No severe complications (CD ≥ 3) were observed.

No 90-days mortality was recorded.

No positive resection margins were observed in the two series, but one tumor capsule rupture occurred in LS requiring conversion to open.

The final histopathological findings of the surgical specimens are presented in Table 1.

Accordingly to Fletcher et al. [13], the majority of the GISTs were classified as very low and low risk (83.9%), while 8 patients (9.9%) presented a medium risk and 5 were high risk cases (6.2%), with no significant difference between the two series.

### Long-Term Oncological Outcome

Eventually, nine patients lost soon after surgery were never followed up and were excluded, thus 72 patients (88.9%) were enrolled for the long-term analysis. Mean and median follow-up time were 47.4 months and 39.8 months, respectively (range 4.1–135.6 months).

Five patients (6.9%) experienced recurrence, all of them were treated with Imatinib. Only one stopped therapy due to adverse effects, and one patient underwent surgery to remove a caval lymph node. All patients are still alive.

Overall DFS at 24, 36, and 60 months was 95.1%, 92.9%, and 90.6%, respectively (Table 3).

In the univariate Cox analysis, the factors statistically associated with shorter DFS were mitotic rate, tumor size, history of GI bleeding, ulcerated lesion, and Fletcher’s classification.

We performed two separate multivariate Cox regression models. In the first we entered GI bleeding, ulcerated lesion, and Fletcher’s risk class as factors, while in the second the Fletcher’s criteria were split into tumor size and mitotic rate. Both models indicated the Fletcher’s criteria as the only factors associated with increased risk of recurrence either in combination (HR 5.2358; 95% CI 1.6890–16.2301; *p* = 0.004), or as single indicator (tumor size: HR 2.0927; 95% CI 1.278–3.5693; *p* = 0.008, and mitotic rate: HR 7.3453; 95% CI 1.5891–33.9519; *p* = 0.011) (Figure 9, Figure 10, Figure 11 and Figure 12).

The Kaplan-Meier curves demonstrated significantly shorter 5-year DFS in the patients with ulcerated lesion (58.3% vs. 96.2%; *p* = 0.0008), mitotic rate >5/50 HPFs (64.3% vs. 93.6%; *p* = 0.0204), tumor size > 5 cm (45.0% vs. 97.4%; *p* = 0.0078), and Fletcher’s high and intermediate risk class (50.0% and 52.6% vs. 100.0%; *p* < 0.0001), whereas the DFS difference in patients presented GI bleeding was almost significant (76.3% vs. 95.6%; *p* = 0.0560) (Figure 9, Figure 10, Figure 11 and Figure 12).

In Figure 13 is depicted the overall disease-free survival.

A summary of peri-operative data from the most relevant publications about robotic GGs resections is depicted in Table 4.

## 4. Discussion

Surgery is the only potential cure for primary localized GGs to prevent disease progression [40]. Neo-adjuvant should be considered for locally advanced and metastatic diseases; a proper pre-operative work-up and a multidisciplinary approach is essential to deal with these tumors, which should be managed in experienced centers [41].

The risk of recurrence of GGs is lower than in other location of gastro-intestinal tract (e.g., small intestine) [8,42,43,44]. Interestingly, the size >10 cm, mitotic index (>5 mitoses per 50 HPFs high-power field), and KIT/PDGFR mutations are well-known risk factors for recurrence [45,46]. This must be taken into account to classify patients into recurrence risk groups: very low/low risk, intermediate risk, or high-risk, as proposed by the American Joint Committee on Cancer (AJCC) [47,48,49,50,51]. The diffusion of laparoscopic approach for GGs resections was slow and steady, also considering the rarity of the disease. The 2007 NCCN guidelines suggested laparoscopic indication only for <2 cm diameter tumors [52]; this indication was later brought to 5 cm only for lesions limited to the anterior gastric wall [53,54,55,56,57]. Nowadays, according to the latest guideline, GISTs tumors less than 2 cm in diameter are generally closely monitored [55], and tumors larger than 10 cm might be considered suitable for a minimally invasive approach in experienced centers [52].

On the other hand, the real benefits of MIS (minimally invasive surgery) approaches in very large lesions should be consider carefully, because in such cases an abdominal incision will be necessary anyway to remove the specimen [18,55,58,59,60,61].

Laparoscopic partial or total gastrectomy may be indicated, especially for large iuxtacardial or distal tumor locations. Those procedures require experienced and skilled surgeons in advanced laparoscopic surgery, mostly in performing intracorporeal anastomosis. The MIS approach should always be considered, knowing its benefits [62,63].

The first robotic series of five GGs treated by robotic approach was reported by Buchs et al., concluding that the approach was safe and effective without recurrences [32].

In 2016 Vicente E. et al. reported on three patients who underwent robotic gastric surgery for GISTs, highlighting how particular unfavorable locations might be better managed with the robotic approach, although the reported operative time was higher compared to open and laparoscopic procedures [19].

The authors described a new surgical cooperation between robotic and endoscopic surgery, where the GG was identified through the endoscopic view and, after a submucosal injection, was dissected through robotic surgery (Table 4) [36].

Zhao et al. published their series of 11 patients operated for GGs located at cardial and subcardial regions, by robotic approach using a completely intracorporeal technique for suturing after excision (Table 4) [35].

In a recent publication Solaini et al. compared 101 consecutive gastric wedge resections for GIST between 2009 and 2019, (14 open, 63 laparoscopic, and 24 robotic) from five Italian centers. The robotic approach showed longer operative time (robotic 180 min vs. laparoscopic 100 vs. open 110; *p* < 0.0001) [38]. The period after the surgery and before the first flatus and the length of the hospital stay were significantly longer in the open series [38]. Complication rates were similar among the groups. Focusing on gastric suture, the robotic handsewn suture performed in 19 over 24 cases, compared to the laparoscopic one (performed in 20 cases over 63), showed that the operative time was longer with robotics (*p* = 0.007).

In Table 4 we have reported a review of our patients, tumors, and peri-operative characteristics compared to other robotic cases series resections for GISTs published so far. The whole number of resections retrieved in literature was 153. Despite peri-operative outcomes like operative time, blood loss, and complications rate, R0 resections are similar to other reports in literature (Table 4). We treated with a robotic approach more challenging patients affected by moderate or high-risk tumors in unfavorable positions, often performing simultaneous procedures or major gastric resection, without negative impact on outcomes. Moreover, we performed a longer than a three year follow-up to better define long-term safety and effectiveness of the robotic approach.

The two minimally invasive groups of patients did not differ statistically, except for the tumor location and tumor diameter > 5 cm was more frequent in the RGR group. However, in the robotic group we found larger lesion resections located in a more complex surgical field.

As already discussed by other authors, MIS approaches might be considered safe and effective, considering the available literature. Laparoscopy should be taken into account in patients with smaller tumors in more favorable positions, while robotic surgery may be more effective for larger tumors in more difficult surgical fields. To the date, due to the small number of retrospective studies on robotic approach to GGS, it is not possible to draw definitive conclusions [64,65].

In order to overcome limits of MI techniques like laparoscopic and endoscopic surgery, some authors described laparoscopic and endoscopic cooperative surgery (LECS) [66]. The drawbacks of LECS are considered the approach to only small lesions and the potential risk of gastric tumor cells spilling into the abdomen because during the procedures the gastric wall may be opened [67,68]. Some modifications to LECS were described, like inverted LECS [69], non-exposure technique (CLEAN-NET) [70], endoscopic full-thickness resection [71], and closed-LECS [72,73]. Despite the fact that there are many reports on LECS procedures which affirm the safety and efficacy of the technique, to date there is not clear evidence to perform GGs resection using LECS. Present literature has only demonstrated that LECS operation is a good treatment option for small mesenchymal cancers [74].

As stated above, LECS should be performed only to treat small tumors which have a high risk of tumor spilling. On the contrary, RS should be dedicated to more challenging procedures, and for that reason we think that indications are the main difference between the two techniques. The careful case by case multimodal discussion could help centers to perform the best procedures for each case in the absence of clear literature indications and guidelines based on prospective studies and randomized controlled trials.

Robotic approaches seem to be longer than laparoscopy, but it should be considered that the use of staplers significantly reduces the operating time of LS. The handsewn robotic technique for gastric reconstruction, compared with the stapler technique, showed some advantages: it is cheaper; allows a regulated and tailored resection avoiding large unnecessary gastric wall removal, which is more appropriate for challenging areas; and moreover represents an optimal training model for young surgeons.

The differences in operative time due to the use of staplers during LS and hand sewed suture during RS do not allow a general comparison. The costs the robotic procedures are estimated to be 21% more expensive than laparoscopy [33]. The extra-time required for docking and robotic set up of devices generally decreases with experience of dedicated surgeons teams.

No-touch technique, respect for margins, and reduction risks of rupture may result in an expansion of indications, a lower conversion rate, and higher possibility of organ-sparing surgery. Furthermore, as already reported for Hepato-Pancreatico-Biliary (HPB) surgery, the robotic approach may synthesize advantages of the MIS approach for a safe and optimal surgical technique. Considering wedge gastric resections as low-middle difficult procedures, a proper hub and spoke learning program between surgical units could reduce the patients’ mobility, achieving the best standard of care without the need of reaching referral centers [75].

To our knowledge only a few reports on robotic approaches for gastric GISTs were published, probably due to the low diffusion of robotic devices combined with low tumor incidence. Giulianotti and Hashizume in 2003 reported first robot-assisted gastrectomies in patients affected by adenocarcinoma [76,77]. Considering the low tumor incidence and the evident lack of literature on MIS indications to GGs, we think that sharing multicentric huge casuistry on the topic might encourage teams to report their experiences and it may improve knowledge, allowing even more evidence to best pose indications to GGs MIS. In our report we tried to clearly define well-established prognostic factors and confirm through a wider casuistry the already reported associations.

Our study has some limitations.

The majority of larger tumors were approached with robotics, and due to the retrospective nature of the study a selection bias for the comparative study cannot be excluded.

During the study period technology, surgical experience, as well as guidelines, changed.

As discussed above, as indications changed in the last ten years, the same type of patients excluded at the beginning of the experiment, due to tumor size or lack of robotic platforms, may have been treated differently in the last period.

## 5. Conclusions

Robotic surgery, when available, allows a safer and easier minimally invasive treatment for GGs larger than 5 cm and located in unfavorable surgical fields (iuxtacardial, lesser curvature, and antro-pyloric regions).

Robotic-platform offers the advantage of “tailored-resections” and, also in difficult conditions, the possibility of performing an “organ-sparing” approach. This prevents the postoperative stomach deformations or strictures due to the straight linear resection coming from the use of conventional laparoscopic staplers.

Finally, since robotic technologies have higher costs, their use should be reserved at tertiary referral centers for the more challenging cases, including obese patients or when associated complex surgical procedures are required.

Considering the low incidence of GG, further multicentric prospective studies are necessary to better define the results of our observations.

## Figures and Tables

**Figure 1 cancers-13-04351-f001:**
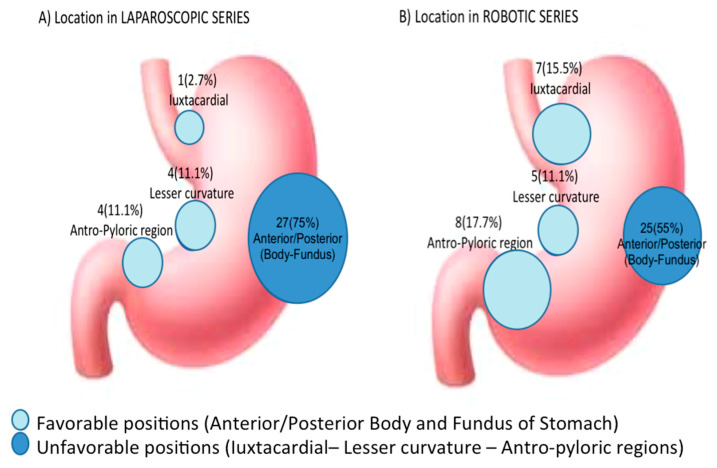
Favorable and unfavorable position of gastric GIST tumors.

**Figure 2 cancers-13-04351-f002:**
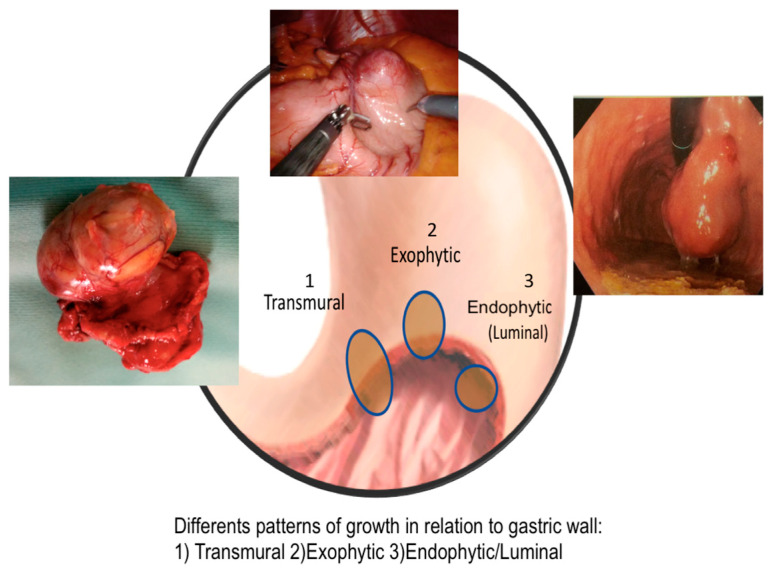
Different pattern of GIST growth.

**Figure 3 cancers-13-04351-f003:**
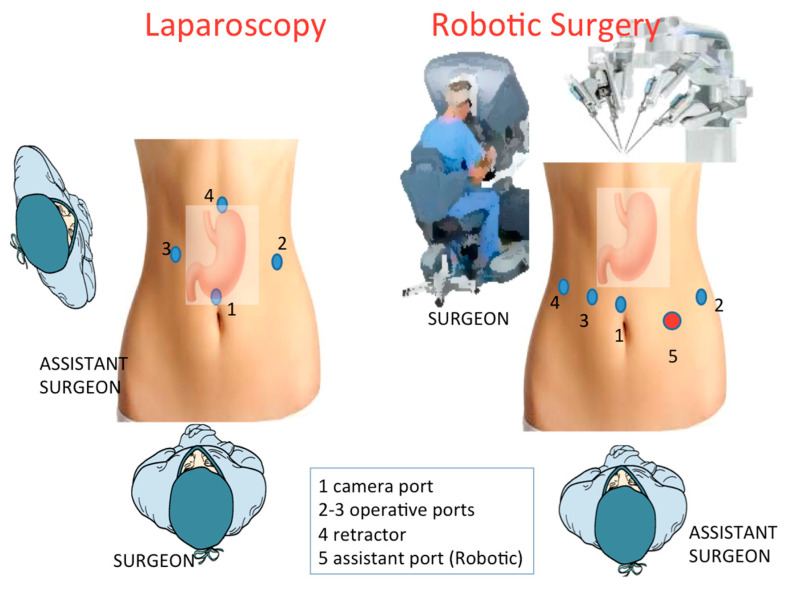
Trocar localization in laparoscopic and robotic approaches.

**Figure 4 cancers-13-04351-f004:**
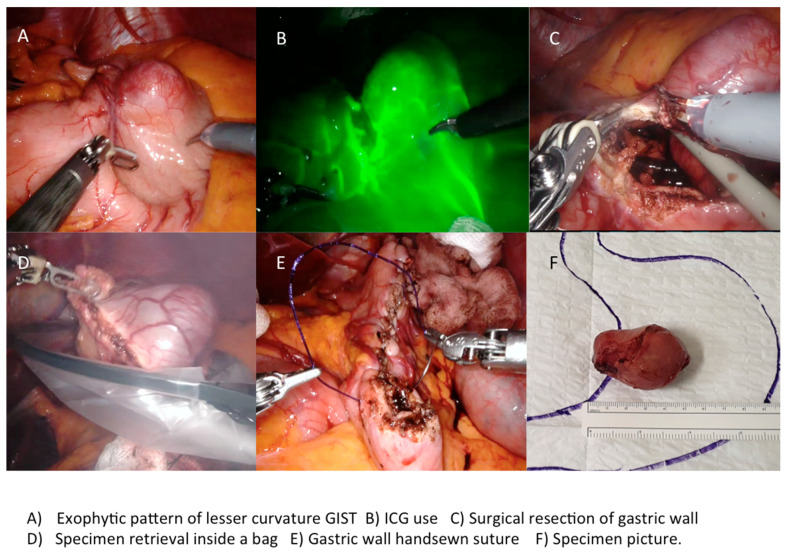
Use of Indocyanine green (ICG) technique during surgical procedures and GIST resection.

**Figure 5 cancers-13-04351-f005:**
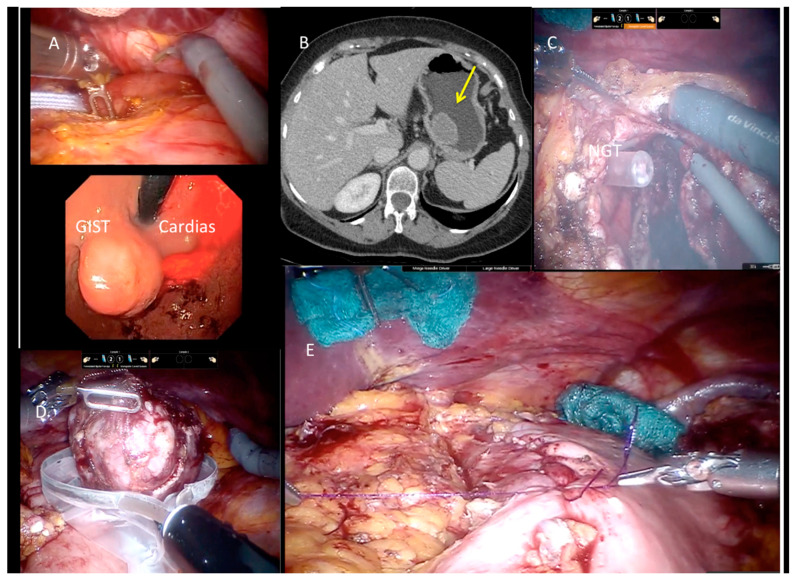
Lesion in CT, endoscopy, and GIST resection.

**Figure 6 cancers-13-04351-f006:**
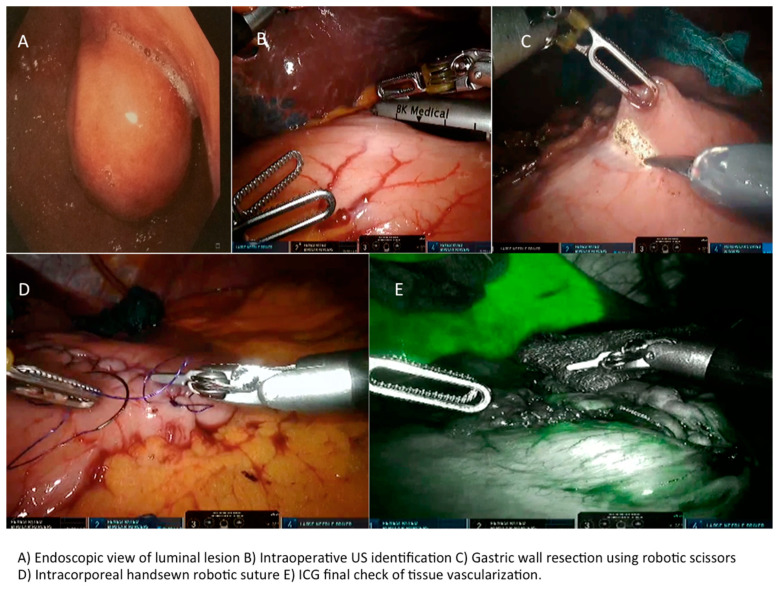
Endoscopic view of luminal lesion and robotic approach.

**Figure 7 cancers-13-04351-f007:**
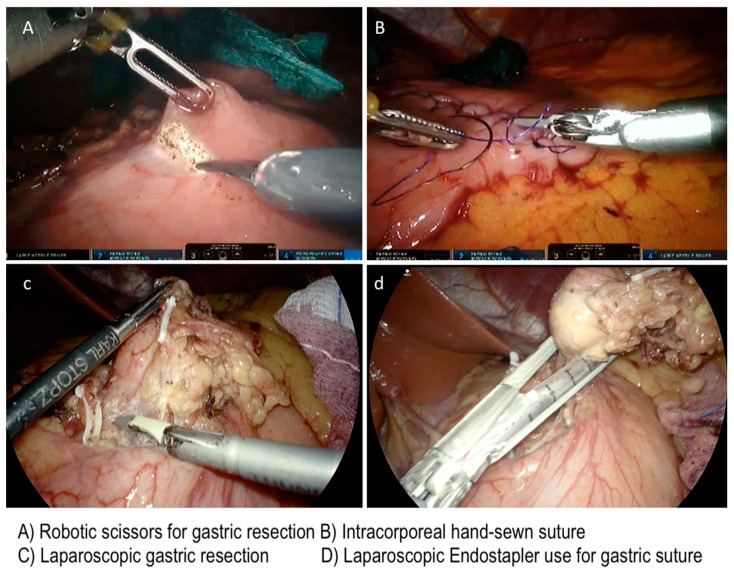
Laparoscopic and robotic approach during resection of gastric GIST.

**Figure 8 cancers-13-04351-f008:**
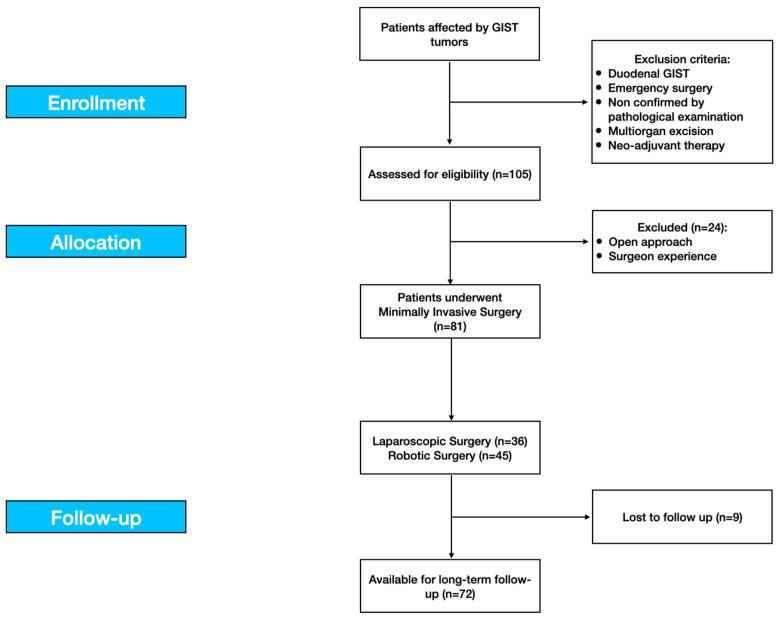
The study flow-chart according to the STROBE statements.

**Figure 9 cancers-13-04351-f009:**
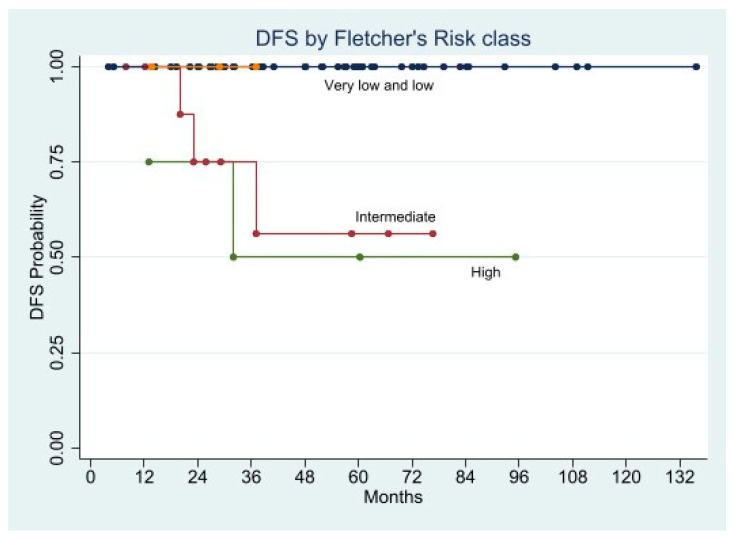
Disease Free Survival by Fletcher’s Risk class.

**Figure 10 cancers-13-04351-f010:**
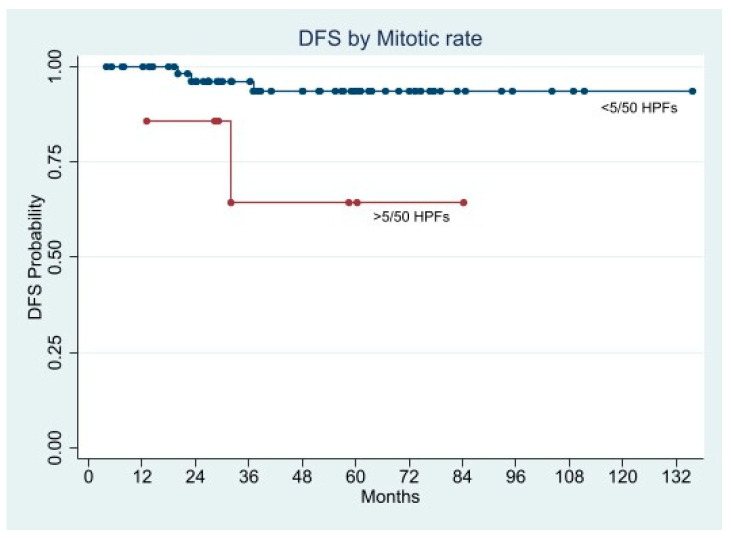
Disease Free Survival by Mitotic rate.

**Figure 11 cancers-13-04351-f011:**
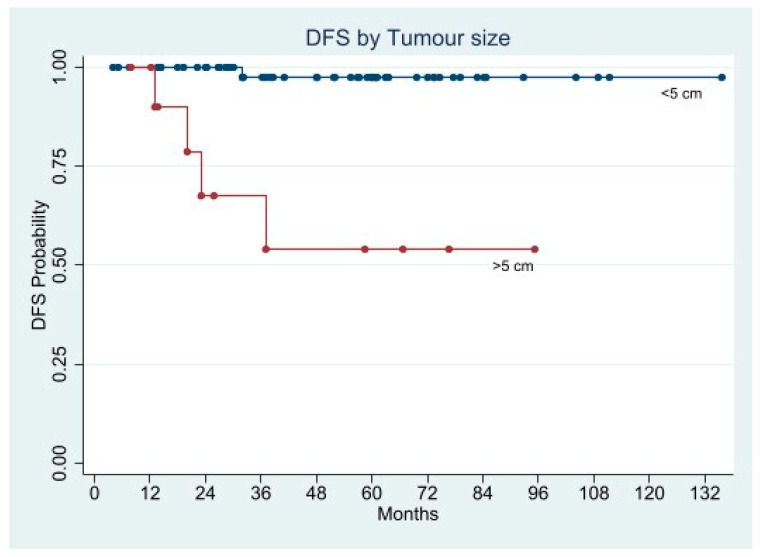
Disease Free Survival by Tumor size.

**Figure 12 cancers-13-04351-f012:**
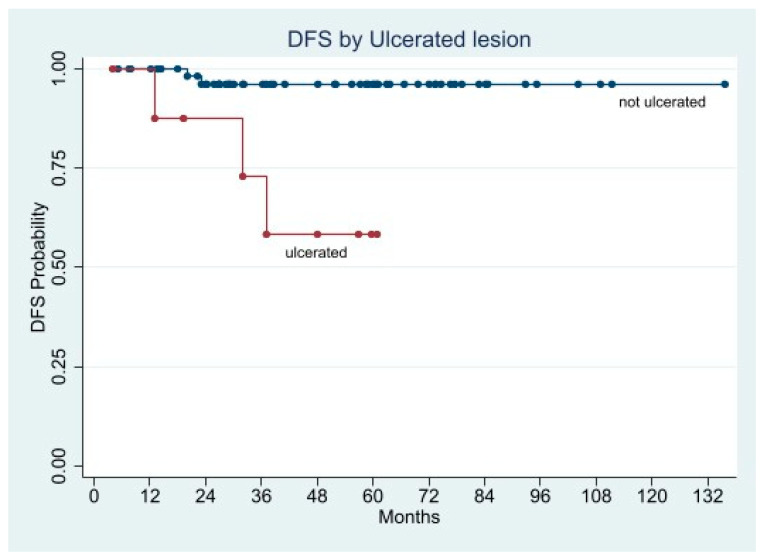
Disease Free Survival by Ulcerated lesion.

**Figure 13 cancers-13-04351-f013:**
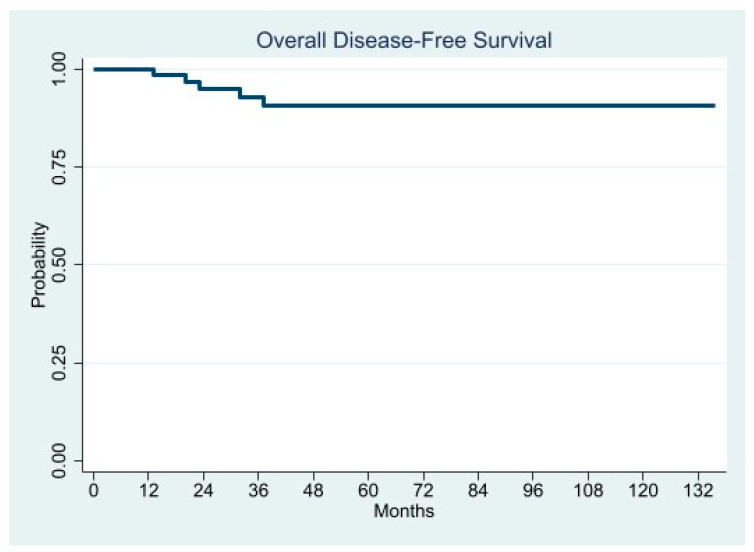
Overall Disease Free Survival.

**Table 1 cancers-13-04351-t001:** Pathologic features.

Variable	Tot	Robotic	Laparoscopy	*p*-Value
		45 (55.5)	36 (44.5)	
Mitotic rate (50 HPF) >5, *n* (%)		5 (11.1)	3 (8.3)	0.7271
Intraoperative tumor rupture, *n* (%)		45 (100)	35 (97.2)	NA
Free margins/R0 resections, (%)		(100)	(100)	1.0000
Immunohistochemistry (pos)				
CD117, *n* (%)		44 (97.7)	36 (100)	1.0000
CD34, *n* (%)		45 (100)	36 (100)	1.0000
DOG-1, *n* (%)		40 (88.8)	33 (91.6)	0.7271
S-100, *n* (%)		3 (6.6)	2 (5.5)	1.0000
Fletcher classification, *n* (%)				
Very low/Low risk	68 (83.9)	36 (80)	32 (88.8)	0.3669
Intermediate risk	8 (9.9)	6 (13.4)	2 (5.6)	0.2896
High risk	5 (6.2)	3 (6.6)	2 (5.6)	1.0000

CD34: Cluster of differentiation molecules 34; CD117: Cluster of differentiation molecules 117 or C-Kit; DOG-1: delay of germination-1; S-100: acidic Ca^2+^-binding proteins.

**Table 2 cancers-13-04351-t002:** Demographics and clinico-pathologic data of gastric GIST treated in the two series Robotic surgery (RS) and Laparoscopy (LS).

Characteristics	Total	Robotic	Laparoscopy	*p*-Value
Total (*n*%)	81 (100)	45 (55.5)	36 (44.5)	
Mean age, (range)	66.6 (35–87)	68.3 (38–87)	64.3 (35–84)	0.1767
Age > 80 years *n* (%)	12 (14.8)	9/45 (20)	3/36 (8.3)	0.7467
Sex, male *n* (%)	35 (43.2)	19 (42.2)	16 (44.4)	0.9800
Body mass index (kg/m^2^) ≥30, *n* (%)	9 (11,1)	6 (13.3)	3 (8.3)	0.3070
ASA classification 3–4, *n* (%)	10 (12,3)	6 (13.3)	4 (11.1)	0.5141
Main clinical manifestations, *n* (%) °				
GI bleeding/anemia	23 (28.4)	14 (31.1)	9 (25)	0.7202
Abdominal pain/discomfort	32 (39.5)	21 (46.6)	11 (30.5)	0.2131
Radiological/diagnostic exams, *n* (%) °				
Endoscopy with biopsy	81 (100)	45	36	1.0000
CT scan (with contrast enhanced)	79 (97.5)	44/45 (97.7)	35/36 (97.2)	1.0000
Abdominal ultrasonography US	81 (100)	45/45 (100)	36/36 (100)	1.0000
US-Endoscopy with biopsy	19 (23.4)	12/45 (26.6)	7/36 (19.4)	0.5792
MRI	7 (8.6)	4/45 (8.8)	3/36 (8.3)	1.0000
Comorbidities presence, *n* (%)	33 (40.75)	20 (44.4)	13 (36.1)	0.5955
Tumor size * (cm) mean (range)	4.4 (1.5–12)	5.1 (1.5–12)	3,7 (1.5–10)	0.0078
size 1–5 cm, *n*	63 (77.8)	30 (66.7)	33 (91.7)	
size ≥ 5 cm, *n*(%)	18 (22.2)	15 (33.3)	3 (8.3)	
Unfavorable gastric location, *n* (%)	29 (35.8)	20 (44.4)	9 (25.0)	0.1140
Cardia/Juxtacardial	8	7 (15.5)	1 (2.7)	0.0701
Lesser curvature	9	5 (11.1)	4 (11.1)	1.0000
Antro-pyloric region	12	8 (17.7)	4 (11.1)	0.5338
Type of growth, *n* (%)				0.8561
Endophytic (luminal)	29 (35.8)	17 (37.7)	12 (33.3)	
Exophytic and Transmural	52 (64.2)	28 (62.3)	24 (66.7)	

* 2 cases <2 cm in each series were incidental findings during other surgical procedures. ° In 2 patients there was an obesity surgery.

**Table 3 cancers-13-04351-t003:** Operative and perioperative outcomes in the two series.

	Tot	Robotic	Laparoscopy	*p*-Value
	81	45 (55.5%)	36 (44.5%)	
Type of gastric resection				0.6511
Wedge resections, *n*/tot (%)	76 (93.8)	43/45 (95.5)	33/36 (91.6)	
Major resections (Gastrectomy), *n*/tot (%)		2/45 (4.4)	3/36 (8.3)	
Conversion to open, *n* (%)	5 (6.2)	2 (4.4)	3 (8.3) °	0.6511
Associated abdominal surgery, *n* (%)		23 (51.1)	13 (36.1)	0.2606
major (colon, liver, hiatal hernia, obesity)	15 (18.5)	10 (22.2)	5 (13.8)	0.3985
minor (cholecystectomy, adhesions)	21 (25.9)	13 (28.8)	8 (22.2)	
Operation time (min), median (range) °°		151 (75–300)	97 (35–185)	0.0002
Effective time of resection/suture °°°		65 (33–115)	49 (25–110)	0.0006
Stapler use for resection, *n* (%)		2 (4.4)	35 (97.2)	<0.0001
Intraoperative endoscopy, *n* (%)	31 (38.3)	17 (37.7)	14 (38.9)	0.8981
Intraoperative ICG use, *n* (%)	15	15 (33.3)	0	
Intraoperative US, *n* (%)	5 (6.2)	3	2	1.0000
Estimated blood loss > 50 mL, *n* (%)		7 (15.5)	5 (13.8)	1.0000
Perioperative transfusion request *n*° cases		1	0	NA
Complications Clavien-Dindo 3–4/Reoperations		0	0	NA
Time to return to bowel function (mean days)		3.1	3.4	0.1543
Time to oral liquid intake (days)		2.9	3.3	0.5484
Postop. hospital length of stay, median (range)		6.7(4–12)	6.3(3–9)	0.3922
Post-operative follow-up, n.ro of patients (%)	72 (88.9)			
- Mean follow-up (months)	47.4			
- Median follow-up (months)	39.8			
Overall Disease-Free Survival (%)				
- 24 months	95.1			
- 36 months	92.9			
- 60 months	90.6			

* Patients with preoperative anemia were already treated. ° In one case the operation was concluded after accessory incision to complete anastomosis after gastrectomy. °° The value includes associated procedures too, when performed. °°° To consider as the effective time of resection and suture (with stapler or handsewn technique).

**Table 4 cancers-13-04351-t004:** Studies on robotic gastro-intestinal stromal tumor (GIST) resections: study, patient, tumor, and peri-operative characteristics.

Author	Years	Type of Study	Period, year	N° Patients	Approach	Gender M;F	Age	BMI, kg/m^2^	Tumor Size, cm	Fletcher Criteria	Resection Type	Tumor Location	Associated Procedures	Conversion, N (%)	Operative Time, min	R0 Resections	Blood Loss, mL	Length of Hospital Stay, days	Follow-Up Time, months	Clavien-Dindo >2	Morbidity	Mortality	Costs
Buchs N. et al. [32]	2010	Retrospective single center	2006–2009	5	Robot	3; 2	Mdn 39 (Ra 32–74)	NA	Mdn 5.5	1 L, 2 I, 2 H	4 WR /1 TG	2 cardia/ 3 distal antrum	No associated procedures	1 (20)	Mdn 192 (Ra 132–285)	5	NA	Mdn 7 (Ra 5–10)	Mdn 18 (Ra 11–27)	0	0	0	NA
Desiderio J. et al. [20]	2013	Retrospective single center	2011–2012	5	Robot	2; 3	M 63.6 (Ra 43–76)	NA	M 5 (Ra 4–7)	2 L, 3 I	5 DG	2 antrum/ 3 prepyloric	No associated procedures	0	Mdn 240 (Ra 210–300)	5	M 96 (Ra 80–120)	M 4.2 (Ra 3–5)	Mdn 13,5 (Ra 12–15)	0	0	0	NA
Vicente et al. [19]	2015	Retrospective single center	2012–2014	6 (3) *	Robot	0; 3	M 55.7 (Ra 41–67)	NA	M 4 (Ra 3–5.5)	2 L, 1 H	1 WR/2 STG	1 cardia/ 1 antrum/ 1 body	Intraoperative endoscopy	0	M 333.3 (Ra 230–540)	3	NA	M 7.7 (Ra 7–9)	M 16.7 (Ra 8–26)	0	0	0	NA
De Angelis et al. [33]	2017	Retrospective single center	2012–2015	12	Robot	8; 4	Mdn 62.5 (Ra 32–86)	Mdn 23.25 (Ra 21.6–28.4)	Mdn 7.25 (Ra 5.5–11.5)	7 L, 3 I, 2 H	11 WR/1 DG	6 fundus (4 anterior, 2 posterior)/ 1 pylorus/ 2 posterior body/ 3 anterior body	3 intraoperative laparoscopic ultrasound	0	162.5 (Ra 140–220)	12	42 (Ra 25–80)	4 (Ra 3–7)	16 (Ra 5–32)	0	1 wound abscess	1	> 21.6% robotic
Al-Thani H. et al. [34]	2016	Retrospective single center	2009–2010	4	Robot	3; 1	M 45 (Ra 33–67)	NA	M 6 (Ra 3.5–10)	3 L, 1 H	4 WR	4 posterior	1 Intraoperative endoscopy	0	Mdn 360	4	NA	M 8 (Ra 5–8)	Mdn 44 (Ra 0–73)	0	0	0	Cost unfavorable to robot assisted gastrectomy
Arseneaux M et al. [30]	2018	Retrospective single center	NA	3	Robot	2; 1	M 68.7 (Ra 59–78)	NA	M 4.8 (Ra 2.8–7)	NA	3 WR	1 cardia/ 1 antrum/ 1 posterior body	3 Intraoperative endoscopy, 1 Nissen fundoplication, 1 gastric diverticulum dissection, 2 hiatal hernia repair	0	NA	3	NA	M 1.5 (Ra 1–2)	M 3.3 (Ra 1–5)	0	1 acute blood loss anemia	0	NA
Zhao J. Et al [35]	2018	Retrospective single center	2014–2016	11	Robot	5; 6	M 59.5 (Ra 43–77)	M 22.1 (Ra 18.2–26.6)	M 5.3 (Ra 3–7.5)	6 L, 4 I, 1 H	11 WR	11 cardia/subcardial	No associated procedures	0	M 82.7 (Ra 60–110)	11	M 30 (Ra 5–50)	M 3.3 (Ra 2–5)	M 25.5 (Ra 8–40)	0	0	0	NA
Maggioni C. et al. [25]	2019	Retrospective single center	NA	6 (5) **	Robot	2; 3	M 77,6 (Ra 58–87)	NA	M 6,3 (Ra 3.5–8.4)	3 L, 2 H	5 WR	1 lesser curvature/ 1 fundus/ 2 posterior body/ 1 anterior body	No associated procedures	0	M 173 (SD ± 39)	6	NA	M 3 (SD ± 1)	M 12	0	0	0	NA
Shi F. et al. [36]	2019	Retrospective single center	2018–2019	20	Robot	7; 13	M 54.5 (Ra 37–80)	M 22.3 (Ra 19.5–25.2)	M 3.3 (Ra 2.4–5.0)	19 L, 1 I	20 WR	2 cardias/ 2 pyloric/ 5 anterior body/ 4 posterior body/ 1 lesser curvature/ 6 greater curvature	No associted procedures	0	Mdn 115 (Ra 90–160)	20	Mdn 20 (Ra 5–100)	Mdn 6 (Ra 4–10)	Mdn 10 (Ra 3–15)	0	1 pneomonia	0	Mdn 7793.25 (Ra 7128.8–11880.7) hospitalization expensive
Furbetta N. et al. [37]	2019	Retrospective single center	2010–2017	12	Robot	5; 7	M 67.4 (SD ± 2.7)	M 24.9 (SD ± 7.1)	M 3.8 (SD ± 1.4)	8 L, 3 I, 1 H	12 WR	4 greater curvature/fundus/ 1 cardia/ 2 antrum/ 3 lesser curvature/ 2 posterior body	1 liver biopsy, 1 cholecystectomy	0	M 149 (SD ± 16.6)	12	NA	M 4.8 (SD ± 1.1)	M 38.5 (Ra 3–90)	0	1 atrial fibrillation	0	NA
Solaini L. et al. [38]	2019	Retrospective multi center	2009–2019	24	Robot	13; 11	4 ≥ 75, 20 <75	15 ( < 30), 3 ( ≥30), 6NA	Mdn 4.0 (Ra 1.8–7)	19 L, 3 I, 2 NA	24 WR	2 cardias/ 4 fundus/ 2 greater curvature/ 9 lesser curvature/ 3 anterior body/ 4 posterior body	No associted procedures	1 (4.2)	Mdn 180 (Ra 70–390)	23	≤100	Mdn 6 (Ra 2–12)	Mdn 24 (Ra 1–87)	0	0	0	higher cost
Winder A. et al. [39]	2020	Retrospective single center	2017–2019	12	Robot	6;6	Mdn 62 (Ra 43–79)	Mdn 27.5 (Ra 20.3–35.1)	M 4.6 (Ra 2,5–8.3)	9 L, 2 I, 1 H	12 WR	2 anterior body/ 1 greater curvature/ 3 antrum/ 2 posterior body/ 2 fundus/ 1 lesser curvature/ 1 gastrosplenic lig.	1 splenectomy	1 (8.3)	M 192 (Ra 95–250)	12	<50	M 2.7 (Ra 2–6)	NA	0	0	0	NA
Ceccarelli G. et al.	2021	Retrospective multi center	2010–2020	45	Robot	19;26	M 68.3 (Ra 38–87)	6 ( >30)	M 5.1 (Ra 1.5–12)	36 L, 6 I, 3 H	43 WR, 2 MG	7 cardias, 5 lesser curvature, 8 antro pyloric, 25 favorable gastric location	17 intraoperative endoscopy, 15 intraoperative ICG, 3 intraoperative US, 10 major (colon, liver, hiatal hernia, obesity, etc.), 13 minor (cholecystectomy, adhesions)	2 (4.4)	Mdn 151 (Ra 75–300)	45	7 pz >50	Mdn 6.7 (Ra 4–12)	Mdn 39.8 (Ra4.1–135.69)	0	0	0	higher cost

NA not available; * 3 duodenal GIST; ** 1 Schwannoma; L: low risk; I: intermediate risk; H: high risk; WR: wedge resection; TG: total gastrectomy; DG: distal gastrectomy; STG: subtotal gastrectomy; MG: major gastrectomy; M: mean; Mdn: median; Ra: range; SD: standard deviation.

## Data Availability

Data available on request due to restrictions, e.g., privacy or ethical. The data presented in this study are available on request from the corresponding author. The data are not publicly available due to the presence of other sensitive information not included in the study.

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
