# Peer review of "Minimally Invasive Approach to Gastric GISTs: Analysis of a Multicenter Robotic and Laparoscopic Experience with Literature Review"

_cancers, 2021, doi:10.3390/cancers13174351_

Round 1
Reviewer 1 Report
The authors investigated the feasibility and safety of minimally invasive approach for Gastrointestinal stromal tumors (GISTs) and Robotic Resection (RS) comparing to Laparoscopic Surgery (LS). The focus of this paper is very interesting, and they demonstrated the excellent result. The manuscript was clearly summarized. Although I read your paper with interest, the findings are limited by several important factors. The followings should be taken into consideration.
- What are the indications for RS and LS surgery for GISTs respectively, and what are the criteria for choosing one over the other?
- The difference in operative time is due to the difference between hand-sewn sutures and staple sutures, so it may not be possible to make a general comparison.
- Laparoscopic Endoscopic Cooperative Surgery (LECS) is one of the minimally invasive treatments for GISTs, what are the advantages of this technique compared to LECS? Please describe in the discussion section.
- I don't think the results and content of this article are very different from the results and content of other articles, but what are the new findings in this article that are not found in other papers?
- The table is very difficult to understand. Please change it to be more concise and clear.
Author Response
Dear Reviewer,
thank you for considering our paper for publication in your eminent journal.
Please find hereby enclosed our corrections in order to answer to Reviewer’s comments on the manuscript entitled
“Minimally invasive approach to gastric GISTs: Analysis of a multicenter robotic and laparoscopic experience with literature review.”
Graziano Ceccarelli, Gianluca Costa, Michele De Rosa, Massimo Codacci Pisanell5, Barbara Frezza, Marco De Prizio, Ilaria Bravi, Andrea Scacchi, Gaetano Gallo, Bruno Amato, Walter Bugiantella, Piergiorgio Tacchi, Alberto Bartoli, Alberto Patriti, Micaela Cappuccio, Klara Komici, Lorenzo Mariani, Pasquale Avella and Aldo Rocca
that we submit for publication in Cancers.
We have reviewed our paper according to Editor suggestion and we summarized below point by point the analysis made:
Reviewer 1
- What are the indications for RS and LS surgery for GISTs respectively, and what are the criteria for choosing one over the other?
We dedicated a paragraph to advantages and indications LS and RS in the discussion.
- The difference in operative time is due to the difference between hand-sewn sutures and staple sutures, so it may not be possible to make a general comparison.
We modified discussion as suggested by the reviewer.
- Laparoscopic Endoscopic Cooperative Surgery (LECS) is one of the minimally invasive treatments for GISTs, what are the advantages of this technique compared to LECS? Please describe in the discussion section.
As suggested by reviewer, we add a dedicated paragraph to LECS in discussion.
- I don't think the results and content of this article are very different from the results and content of other articles, but what are the new findings in this article that are not found in other papers?
Thank you for your comment. Before limitations we add a dedicated paragraph in discussion to describe our aim.
- The table is very difficult to understand. Please change it to be more concise and clear.
Tables were corrected

Reviewer 2 Report
In this paper the Authors have reviewed the experience of 3 surgical units in minimally invasive surgical resection for gastric GIST. The topic is potentially intriguing and minimally invasive surgery is surely of benefit in most cases of gastric GIST. Also, robotic surgery may help to manage also more technically demanding procedures (unfavorable position, larger tumors, etc.).
Unfortunately, the paper does not add anything to the current knowledge on the topic. The only significant results reported by the Authors are about well-established prognostic factors (i.e. number of mitosis and size) or already reported associations (i.e. longer time of robotic surgery compared with laparoscopic surgery).
Importantly, as stated by the Authors, the retrospective nature of the study is unavoidably associated to significant bias treatment hampering the soundness of the study.
Moreover, although stated in the STROBE flow-chart, inclusion criteria are not completely clear in the text. Indeed, 24 patients were excluded due to positioning (which position?), size (which cut-off has been employed?), open approach or surgeon experience (which criteria has been used to define surgical experience? case-load? role?).
Also, the case series is quite heterogeneous as it has been reported that about 18% of patients had concomitant operations, so it seems that some patients had operations for different disease (obesity, colon, etc.), not for gastric GISTs.
Moreover, although a huge amount of variables and data have been analyzed, in most cases they are not very informative (both from a clinical and scientific point of view). Also, in some cases, data are confusingly or insufficiently reported in the manuscript.
- Pages 2-3: authors claim that a prospectively maintained database was retrospectively reviewed but, on the opposite, they also state that “… Medical charts of patients who underwent … were reviewed. Patients were retrieved using the International Classification of Diseases versions 9 (ICD-9™) [disease codes: 211.1 or 235.2; procedure codes 44.4x to 43.9x and 54.21 and/or 00.39]…” These sentences should be clarified.
- Pag.3: “…Tumour risk recurrence was calculated according to Fletcher and/or Miettenen score [4,8,29] and the data are listed in Table 3…”. They are slightly different classifications and only Fletcher system is reported in table 3. It should be clarified.
- Pag. 3: “…The mitotic index was measured through the HPF and high mitotic index was defined as a number of mitoses >50 HPF…” Please correct the sentence (add number).
- Pag. 4 – Figure 1: the colors to identify favorable vs. unfavorable should be switched.
- Pag. 4 Table 1: Tumor size variable should be probably moved in table 4, as the Authors stated they considered size at pathologic examination.
- Pag. 5 Table1: Two type of growth are showed in the table, but three patterns are reported in the maintext and in figure 2. Need to be clarified.
- Pag.5: “… Follow-up data collection was performed through telephone interview and/or regular outpatient visits, with CT scan and gastroscopy 6 months after surgery.” The follow-up policy adopted by the Authors is not clear. They perform CT and gastroscopy every 6 months for all patients with no difference among the different risk class?
- Pag.5: “…PET-CT was scheduled after eventual chemotherapy or in case of recurrence.” The term chemotherapy should be amended as potentially misleading. Tyrosine kinase inhibitor (or similar) should be preferred.
- Pag.5: Cause-specific survival (CSS) and disease-free survival (DFS) were calculated defining CSS. Why did the authors decide to use CSS if no death for GIST was observed?
- The section on technical notes should be improved as it is quite vague in this form.
- Pag.9 Results: the Authors should more clearly report the actual number (eventually 72) of analyzed patients in the main text.
- Pag. 10. Please see comment on the use of term “chemotherapy”.
- Pag.11: “… All patients showed complications Clavien-Dindo ≤2.” This sentence should be clarified. What does it means? Complication rate was 100%? Or no severe complications (CD>3) were observed?
- Pag.11: “… The majority of the GISTs were classified as very low and low risk (83.9%), while 8 patients (9.9%) presented a medium risk and 5 were high risk cases (6.2%), with no significant difference between the two series [13]…” What’s the meaning for the reference #13 at the end of this sentence?
- Table 2: Reporting “0 complications” and “0 90 days mortality” has probably poor significance in this table.
- Table 3: “Tumor capsule integrity” should be changed in “intraoperative tumor rupture” as they have different biological and pathologic meaning.
- Data on recurrence should be provided.
- Check Y-axis of K-M curves (probably modify scale or %).
- Discussion section should be improved as too vague.
Author Response
Dear Editor,
thank you for considering our paper for publication in your eminent journal.
Please find hereby enclosed our corrections in order to answer to Reviewer’s comments on the manuscript entitled
“Minimally invasive approach to gastric GISTs: Analysis of a multicenter robotic and laparoscopic experience with literature review.”
Graziano Ceccarelli, Gianluca Costa, Michele De Rosa, Massimo Codacci Pisanell5, Barbara Frezza, Marco De Prizio, Ilaria Bravi, Andrea Scacchi, Gaetano Gallo, Bruno Amato, Walter Bugiantella, Piergiorgio Tacchi, Alberto Bartoli, Alberto Patriti, Micaela Cappuccio, Klara Komici, Lorenzo Mariani, Pasquale Avella and Aldo Rocca
that we submit for publication in Cancers.
We have reviewed our paper according to Editor suggestion and we summarized below point by point the analysis made:
Reviewer 2
- Although stated in the STROBE flow-chart, inclusion criteria are not completely clear in the text. Indeed, 24 patients were excluded due to positioning (which position?), size (which cut-off has been employed?), open approach or surgeon experience (which criteria has been used to define surgical experience? case-load? role?).
We specified data as requested. “Positioning” and “Tumor size” were deleted because it was a mistake due to the use of a sample STROBE used in other studies.
- Also, the case series is quite heterogeneous as it has been reported that about 18% of patients had concomitant operations, so it seems that some patients had operations for different disease (obesity, colon, etc.), not for gastric GISTs.
As described in the text a rate of patients underwent surgery for other indications and the diagnosis was incidental due to the asymptomatic presentation of the tumor.
- Pages 2-3: authors claim that a prospectively maintained database was retrospectively reviewed but, on the opposite, they also state that “… Medical charts of patients who underwent … were reviewed. Patients were retrieved using the International Classification of Diseases versions 9 (ICD-9™) [disease codes: 211.1 or 235.2; procedure codes 44.4x to 43.9x and 54.21 and/or 00.39]…” These sentences should be clarified.
The sentence “. Patients were retrieved using the International Classification of Diseases versions 9 (ICD-9™) [disease codes: 211.1 or 235.2; procedure codes 44.4x to 43.9x and 54.21 and/or 00.39]” was deleted. It was used only to check the data.
- 3: “…Tumour risk recurrence was calculated according to Fletcher and/or Miettenen score [4,8,29] and the data are listed in Table 3…”. They are slightly different classifications and only Fletcher system is reported in table 3. It should be clarified.
Thank you for your comment. We deleted the Miettenen classification.
- 3: “…The mitotic index was measured through the HPF and high mitotic index was defined as a number of mitoses >50 HPF…” Please correct the sentence (add number).
In our database we registered patients dividing high and low mitotic index. The cut point was defined as 5 of more mitoses registered at 50 HPF
- 4 – Figure 1: the colors to identify favorable vs. unfavorable should be switched.
Colors were inverted.
- 4 Table 1: Tumor size variable should be probably moved in table 4, as the Authors stated they considered size at pathologic examination.
Tables were corrected as requested.
- 5 Table1: Two type of growth are showed in the table, but three patterns are reported in the main text and in figure 2. Need to be clarified.
The text was corrected as depicted in the table “endoluminal, exophytic or transmural”
- 5: “… Follow-up data collection was performed through telephone interview and/or regular outpatient visits, with CT scan and gastroscopy 6 months after surgery.” The follow-up policy adopted by the Authors is not clear. They perform CT and gastroscopy every 6 months for all patients with no difference among the different risk class?
Follow-up was scheduled according to Italian Guidelines, and it is specified in the text.
- 5: “…PET-CT was scheduled after eventual chemotherapy or in case of recurrence.” The term chemotherapy should be amended as potentially misleading. Tyrosine kinase inhibitor (or similar) should be preferred.
We corrected the text as suggested.
- 5: Cause-specific survival (CSS) and disease-free survival (DFS) were calculated defining CSS. Why did the authors decide to use CSS if no death for GIST was observed?
We changed the method and results sections as suggested.
- The section on technical notes should be improved as it is quite vague in this form.
Our aim concerning technical notes was only devoted to underline the most important advantages and drawbacks of techniques, we think that a more extensive technical description should be address to other type of papers.
Obviously, if you have specific comments we are pleased to receive them and to better review our paper.
- 9 Results: the Authors should more clearly report the actual number (eventually 72) of analyzed patients in the main text.
We reported the clear number of patients included in our study.
- 10. Please see comment on the use of term “chemotherapy”.
We corrected the text as suggested.
- 11: “… All patients showed complications Clavien-Dindo ≤2.” This sentence should be clarified. What does it means? Complication rate was 100%? Or no severe complications (CD>3) were observed?
We corrected the text as suggested.
- 11: “… The majority of the GISTs were classified as very low and low risk (83.9%), while 8 patients (9.9%) presented a medium risk and 5 were high risk cases (6.2%), with no significant difference between the two series [13]…” What’s the meaning for the reference #13 at the end of this sentence?
We clarified the meaning of reference 13.
- Table 2: Reporting “0 complications” and “0 90 days mortality” has probably poor significance in this table.
We correct the table 2 as suggested.
- Table 3: “Tumor capsule integrity” should be changed in “intraoperative tumor rupture” as they have different biological and pathologic meaning.
We correct the text as suggested.
- Data on recurrence should be provided.
We added all the available data on recurrence.
- Check Y-axis of K-M curves (probably modify scale or %).
We changed figures as requested. We used the original STATA output raising %.
- Discussion section should be improved as too vague.
Discussion was improved following Reviewer 1 suggestions.
English was revised with mother-tongue language revision.
Aldo Rocca,
Department of Medicine and Health Sciences "V. Tiberio",
University of Molise, Campobasso, Italy,
Via Francesco De Sanctis, 1, 86100 Campobasso CB,
e-mail: aldo.rocca@unimol.it

Round 2
Reviewer 1 Report
The manuscript has been much improved and is in a nice condition now.
Reviewer 2 Report
The Authors properly replied to most issues; however some crucial points (poor originality, retrospective nature, bias treatment, heterogeneity of the series) cannot be solved. Therefore, the paper is not adequate for publication in a journal as Cancers, in my opinion.
Minor points:
- at present, Miettinen classification is the most employed classification;
- tables format should be significantly improved